# Extraction, Isolation and Nutritional Quality of Coffee Protein

**DOI:** 10.3390/foods11203244

**Published:** 2022-10-17

**Authors:** Rewati Raman Bhattarai, Hayder Al-Ali, Stuart K. Johnson

**Affiliations:** 1School of Molecular and Life Sciences, Faculty of Science and Engineering, Curtin University, Bentley, WA 6102, Australia; 2Ingredients by Design Pty Ltd., Lesmurdie, WA 6076, Australia

**Keywords:** coffee protein, extraction, ultrafiltration, branched-chain amino acids, in vitro protein digestion, polyphenols, antioxidants

## Abstract

Coffee protein is reported to have high levels of branched-chain amino acids of value in sports nutrition and malnutrition recovery. However, data demonstrating this unusual amino acid composition are limited. We investigated the extraction and isolation of protein concentrates from coffee bean fractions, viz. green coffee, roasted coffee, spent coffee and silver skin, and determined their amino acid profile, caffeine content and protein nutritional quality, polyphenol content and antioxidant activity. Alkaline extraction/isoelectric precipitation gave lower concentrate yields and protein content than alkaline extraction/ultrafiltration. The protein concentrate from green coffee beans had a higher protein content than those from roasted coffee, spent coffee and silver skin, regardless of extraction method. The isoelectric precipitated green coffee protein concentrate had the highest in vitro protein digestibility and in vitro protein digestibility corrected amino acid score (PDCAAS). Silver skin protein concentrate had a very low digestibility and in vitro PDCAAS. In contrast to a previous finding, the amino acid levels in all coffee concentrates did not demonstrate high levels of branched-chain amino acids. All protein concentrates had very high levels of polyphenols and high antioxidant activity. The study suggested investigating coffee protein’s techno-functional and sensory attributes to demonstrate their potential applications in different food matrices.

## 1. Introduction

Coffee beans are a globally utilised raw material to produce coffee beverage. The green coffee beans are roasted to give roasted coffee beans and silver skin as the by-product. After grinding, these roasted coffee beans are used to make a coffee beverage with the resultant waste material of spent coffee grounds. All these fractions of coffee beans contain an appreciable amount of protein. For instance, spent coffee grounds are high in protein at ~13.5–19.5 g/100 g dry basis [1] and high in polyphenolic antioxidants [2]. Several publications are suggesting that proteins in coffee beans have an unusual and potentially valuable composition of amino acids. Firstly, coffee beans are reported to have a high percentage of essential amino acids compared to other plant proteins, indicating a high nutritional quality [3]. Secondly, Campos-Vega et al. reported they have a very high proportion of branched-chain (leucine, isoleucine and valine) to total amino acids [3]. Branched-chain amino acids are widely used as a supplement for improved exercise performance and more rapid post-exercise recovery; however, the evidence of their benefits is inconclusive [2]. Coffee protein has also been reported to have a high Fischer ratio (branched-chain amino acids/aromatic amino acids). Such proteins are reported to assist those suffering from malnutrition associated with cancers, burns, trauma, and liver failure and may assist children with chronic or acute diarrhoea or milk protein allergies [3]. Due to these quality attributes, the protein from coffee bean fractions may have unique potential as a supplement or food ingredient. 

Recently, ultrasound-assisted alkaline extraction with acid precipitation has been used to produce protein concentrates from coffee beans silver skin [4]. However, this method, in general, results in (a) loss of proteins in the acid-soluble fraction after the major proteins are precipitated and (b) high salt (sodium) content of the resulting protein concentrate. Ultrafiltration of the alkaline extract is an alternative approach to produce protein concentrate with high yield and content. However, there are no publications reporting the effectiveness of ultrafiltration in the production of coffee concentrates. 

Coffee beans are rich sources of polyphenolic compounds with high antioxidant properties [3] and anti-inflammatory activity [5], such as caffeine and chlorogenic acids [6]. Murthy and Naidu have successfully made polyphenol extracts with high antioxidant activity from coffee by-products (i.e., silver skin, spent coffee ground) [7]. Regazzoni et al. reported high levels of polyphenols and antioxidant activity in the extracts from green coffee beans and their by-products [5]. Prandi et al. reported a higher protein recovery from green coffee beans than from silver skins using protease-assisted extraction [8]. They found increased antioxidant, antityrosinase, and antimicrobial activities of the green coffee protein concentrate than that from silver skins. However, there is no information in the literature on the phenolic content and antioxidant activity of coffee protein concentrates prepared by ultrasound-assisted extraction and ultrafiltration. 

Therefore, this research aimed to characterise the amino acid content, nutrient and polyphenolic composition, antioxidant activity and in vitro protein nutritional quality of coffee protein concentrates from green coffee, roasted coffee, spent coffee and silver skin prepared by ultrasound-assisted alkaline extraction with isoelectric precipitation and ultrasound-assisted alkaline extraction with ultrafiltration.

## 2. Materials and Methods

### 2.1. Materials

The following coffee bean fractions were provided by Protein Unlocked Pty Ltd. (Nedlands, WA, Australia) manufactured from the same whole green coffee beans (La Virgen, Huila, Colombia): green coffee beans, roasted coffee beans, and coffee silver skins. These fractions were ground to Espresso fine using a Breville coffee grinder (Botany, NSW, Australia). Spent coffee grounds were prepared in the laboratory using a coffee filter (low-pressure brewing) and ground roasted coffee beans: boiling water (1:1). The remaining coffee in the filter (spent) was dried using an air oven at 50 °C for 24 h. Soy protein concentrate (70/100 g protein as is, *N* × 6.25) was kindly provided by Archer-Daniels-Midland Company (ADM) (Sydney, Australia) and was used as benchmark protein. 

### 2.2. Methods 

#### 2.2.1. Coffee Protein Concentrates Made by Alkaline Extraction Followed by Isoelectric Precipitation (IP)

The protein concentrate was made from each ground coffee fraction (20 g) using alkaline extraction followed by isoelectric precipitation (IP) based on the method published by Wen et al. [4]. Each protein concentrate was made in duplicate. Protein extraction was conducted using a 1:10 ratio of coffee: water (*w/w*). The pH of the slurry was adjusted to 10 using 1 N NaOH and stirred for 1 h. Then, the mixture was centrifuged (Model 5810 R, Hamburg, Germany) at 4000× *g* for 30 min at 4 °C. The extraction was repeated using the pellet from the first centrifugation. Then, both supernatants were combined. The combined supernatant was adjusted to pH 4.5 using 1N HCl, then centrifuged (Model 5810 R, Hamburg, Germany) at 4000× *g* for 30 min at 4 °C. The resulting protein concentrate pellet was neutralised then freeze-dried (Model ALPHA 1-2 LO, Christ, Osterode am Harz, Germany). The dried concentrate was vacuum packed and stored at 4 °C. 

#### 2.2.2. Moisture, Protein and Caffeine Content Analysis

The moisture content of raw coffee fractions and their protein concentrates prepared by alkaline extraction/isoelectric precipitation was measured in duplicate using the oven drying method at 105 °C to constant weight [9]. The protein content was measured in duplicate using the Kjeldahl digestion and distillation method (Model Kjeltec™ 2100 Distillation unit, Foss, Hillerod, Denmark) using *N* × 6.25 conversion factor [9]. Data were expressed as g/100 g db (dry basis).

Caffeine content of the coffee protein concentrates were analysed in triplicate as per Milić et al. and expressed as g/100 g db [10].

#### 2.2.3. Total Mass Recovery

The coffee protein concentrate total mass recovery was calculated as:Total mass recovery % = (weight of coffee protein concentrate (g as is)/weight of original raw material (g as is)) × 100

#### 2.2.4. Protein Recovery

The protein recovery was calculated as:Protein recovery (%) = (total weight of protein in the coffee protein concentrate (g dry basis)/total weight of protein in raw material (g dry basis) × 100

### 2.3. Green Coffee Protein Concentrate Manufacture Using Alkaline Extraction/Ultrafiltration and the Quality Assessment of the Concentrate

Based on its highest yield of protein concentrate by alkaline extraction/isoelectric precipitation, the green coffee bean was chosen for further investigation of the potential for alkaline extraction/ultrafiltration to improve protein concentrate yield and protein content and to undertake the in-depth investigation of the compositional properties of the protein concentrates.

#### 2.3.1. Manufacture of Green Coffee Protein Concentrate Using Alkaline Extraction/Ultrafiltration (UF)

The alkaline extract of the green coffee was made in duplicate as previously described in Section 2.2.1. It was then ultrafiltered using a 5 kDa membrane on a Minimate tangential flow filtration system (Pall Corporations, Melbourne, Australia). The filtration process was terminated when the solution concentration reached 20% total soluble solids; then, the retentate was neutralised using 1 N HCl. Finally, the retentate was freeze-dried (model ALPHA 1-2 LO, Christ, Osterode am Harz, Germany), vacuum-packed and stored at 4 °C until future use.

The composition, total mass and protein recovery of UF green coffee protein concentrate was determined as previously described in Section 2.2.2, Section 2.2.3 and Section 2.2.4.

#### 2.3.2. In Vitro Protein Digestibility (IVPD) 

The IVPD of the green coffee protein concentrates prepared by alkaline extraction/isoelectric precipitation and by alkaline extraction/ultrafiltration was determined in duplicate following the modified pepsin pancreatin digestion method [11]. Samples of concentrate equivalent to 50 mg protein were incubated at 37 °C with 0.75 mg pepsin (2500 units/mg activity; Chem-Supply, Gillman, SA, Australia) in 7.5 mL of 0.1 N HCl for 3 h. The solution was then neutralised with 3.75 mL of 0.2 N NaOH. Following this, 2 mg pancreatin (Chem-Supply, Gillman, SA, Australia) in 3.75 mL phosphate buffer of pH 8.0 was added, and the sample was incubated for 24 h at 37 °C. The undigested protein in 5 mL of digesta was then precipitated by adding 25 mL of 10% trichloroacetic acid; then, the sample was centrifuged for 30 min at 1000× *g* at room temperature. Nitrogen in the supernatant was determined using the Kjeldahl digestion and distillation method. 

The in vitro protein digestibility (IVPD) was calculated using the equation given below:IVPD % = *N* in total supernatant (g)*/*N* in original sample (g) × 100
where * is corrected for all dilution and subsampling.

#### 2.3.3. Determination of the Amino Acid Profile, Amino Acid Score and In Vitro Protein Digestibility Corrected Amino Acid Score (PDCAAS)

Amino acid profiles of the green coffee protein concentrates were analysed by the ChemCentre (Bentley, WA, Australia) using pre-column derivatisation followed by high-performance liquid chromatography (HPLC) with UV detection [11]. The amino acid content is reported as mg/g protein concentrate (as is).

Amino acid scores were calculated according to FAO/WHO by dividing the amino acid content of the samples (mg/g protein) by the suggested reference pattern of amino acid requirements (mg/g protein) for 1 yr old infants and for adults for nine essential amino acids plus tyrosine and cysteine [12].

The in vitro protein digestibility corrected amino acid score (PDCAAS) of the green coffee protein concentrates were determined by multiplying the IVPD/100 of each sample by the limiting amino acid score (lowest score for an individual essential amino acid [11].

#### 2.3.4. Determination of Polyphenol Content and Antioxidant Activity

Phenolic compounds were extracted in duplicate from the green coffee protein concentrates based on the method described by Wu et al. [4]. In brief, 2 g of coffee protein concentrate was mixed with 15 mL of 80% (*v/v*) aqueous methanol. The mixture was shaken for 2 h in the water bath at 25 °C, then centrifuged at 3220× *g* for 10 min at 4 °C. The supernatant was removed by decantation and saved. This process was repeated twice using the pellet, and the supernatants were combined. The combined supernatant was then evaporated to dryness using a vacuum rotary evaporator. The dry solids were then solubilised in 10 mL methanol and kept at −80 °C under N_2_ in the dark until analysis. 

The total polyphenol content of the extracts was measured in duplicate, according to Zhong et al. [13]. Duplicate 100 µL of polyphenolic extract, gallic acid standards (10–100 mg/L) or 80% (*v/v*) methanol blank was mixed with 200 μL 0.2 N Folin–Ciocalteu phenol reagent and shaken vigorously. After 5 min incubation in the dark environment, 800 μL of Na_2_CO_3_ (5%) was added, and the sample was mixed thoroughly, followed by incubation in the dark for 90 min. Duplicate 200 µL aliquots of the mixture were transferred into a clear 96-well microplate. The absorbance was recorded at 765 nm using a Synergy HT Multi-Detection Microplate Reader (BioTek Instruments, 1266 Inc., Winooski, VT, USA). The results were expressed as mg gallic acid equivalent (GAE)/100 g protein concentrate (as is).

The total antioxidant capacity of the extracts was measured in duplicate using the ABTS method [13]. Duplicate 150 μL aliquots of the polyphenol extracts, Trolox standard or blank, were mixed with 2850 μL 3-ethylbenzothiazoline-6-sulfonic acid diammonium salt (ABTS•+) solutions. The resulting solutions were incubated in the dark for 2 h. The absorbance was recorded at 734 nm. Results were expressed as mg of Trolox equivalents (TE)/100 g protein concentrate (as is).

### 2.4. Statistical Analysis

All data are reported as mean ± standard deviation. The data were statistically analysed using SPSS software (V 26, IBM Corp. Chicago, IL, USA). One-Way ANOVA with Tukey post hoc test or independent sample *t*-tests were performed to compare means, and *p* < 0.05 was considered significant.

## 3. Results and Discussion

### 3.1. Raw Material Proximate Composition and the Protein and Caffeine Contents

The protein content of the different coffee raw materials (green coffee, roasted coffee, spent coffee and silver skin) is given in Figure 1, showing that the roasted coffee had slightly but significantly (*p* < 0.05) higher protein levels than the spent coffee. This indicated that some protein was lost from the roasted coffee bean into the coffee beverage. The green coffee beans and silver skin had slightly but significantly (*p* < 0.05) higher protein levels than the roasted coffee. Our values for the protein content are marginally lower than those reported by Franca et al. [14] for green coffee (17.71 g/100 g db) and Ballesteros et al. [15] for spent coffee (17.44 g/100 g db). However, our values for silver skin and roasted coffee are similar to those reported by Wen et al. [4] (15.53 g/100 g db) and Franca et al. [14] 15.09 g/100 g db), respectively. Based on protein levels alone these finding suggests that any of these raw materials are likely candidates for protein concentrate production. The caffeine contents of the protein concentrates are reported in Table 1, which are at very low levels. Protein concentrates with a low content of caffeine could be attractive to the consumers willing to avoid caffeine in their diet. 

### 3.2. Protein Content of Protein Concentrates Prepared by Alkaline Extraction/Isoelectric Precipitation

Figure 2 shows the protein content of protein concentrates from the different raw materials prepared by alkaline extraction/isoelectric precipitation. The protein concentrates from the green beans had similar (*p* > 0.05) protein content to that from the silver skin but was significantly (*p* < 0.05) higher than that of the protein concentrates from the roasted beans and the spent coffee. The protein content of these concentrates was lower than that reported using the same method on other raw materials such as pea, lentil, and kabuli chickpea (81.7, 78.2 and 63.9 g/100 g db, respectively) [16] and lupin (67.1 g/100 g as is) [17]. In addition, it was lower than the levels in a commercial soy protein concentrate (86.1 g/100 g as is) [18]. Given the low content of the coffee protein concentrates, an alternative method to alkaline extraction/isoelectric precipitation of alkaline extraction/ultrafiltration was investigated.

### 3.3. Total Mass Yields of Protein Concentrates Prepared by Alkaline Extraction/Isoelectric Precipitation

Images of the protein concentrate prepared by alkaline extraction/isoelectric precipitation from 20 g of each raw material are shown in Figure 3, and the total mass yields are presented in Figure 4.

The colour of the concentrates (Figure 3) differed dramatically depending on the raw material used. The silver skin, roasted coffee and coffee spent all gave brownish-coloured concentrate, whereas the green coffee gave a bluish-coloured concentrate. The bluish colour appeared when pH was reduced from 10 (extraction step; the dispersion was dark green at this stage) to 3.5 (precipitation step), and it stayed even during the neutralisation step (pH 7) before drying. This colouration is likely due to polyphenolic compounds that also have antioxidant activity. The compound, cyanidin-3-*O*-rutinoside, is likely responsible for the blue colour of the protein concentrate from the green coffee beans [19,20]. The roast coffee, spent coffee and silver skins concentrates were, however, brown, likely due to degradation or polymerisation of the anthocyanins into melanoidins [21].

The total mass yields (Figure 4) varied greatly (*p* < 0.05) between the samples. That of the green coffee was significantly highest (*p* < 0.05), and that of the silver skin the lowest (*p* < 0.05). The values for the roasted coffee and coffee spent were intermediate. The images of the concentrates from one batch of processing (Figure 3) visually confirm the differences in the concentrate total mass yield.

### 3.4. Protein Yield of Protein Concentrates Prepared by Alkaline Extraction/Isoelectric Precipitation

Figure 5 shows the protein yield of the concentrates prepared by alkaline extraction/isoelectric precipitation. The results follow the same pattern as for the total mass yields (Figure 4). 

The protein yield for the green coffee was by far the highest (*p* < 0.05) and that of the silver skin by far the lowest (*p* < 0.05). The values for the roasted coffee and coffee spent were intermediate. The protein yield from silver skin was lower than that reported at 4.78% using the same method [4]. The meagre yield of protein in the silver skins may be due to the protein being tightly bound in the cellular structure of this seed coating. In the roast coffee and the spent coffee, the proteins may have denatured with reduced solubility because of the roasting process or may have become bound to other cell materials, which reduced their extractability. 

### 3.5. Manufacture of Green Coffee Protein Concentrate Using Ultrafiltration

#### 3.5.1. Protein Content, Total Mass Yield and Protein Yield of Green Coffee Protein Concentrate Prepared by Alkaline Extraction/Ultrafiltration 

The protein content of green coffee protein concentrate prepared alkaline extraction/ultrafiltration (49.41 ± 2.08) was similar (*p* > 0.05) to that prepared by alkaline extraction/isoelectric precipitation (46.00 ± 0.50). This finding contrasts with a previous report showing higher protein content in alkaline extracted/ultrafiltered lupin protein concentrate than in that prepared by alkaline extraction/acid precipitation [17]. Ultrafiltration can purify proteins by washing out low molecular weight molecules such as water, salts and sugars (permeate) and holding back the larger molecular weight proteins by a membrane (retentate). However, our ultrafiltration may not have been extensive enough to wash out all the salts into the permeate. In addition, the soluble dietary fibre may have a high enough molecular weight to be held back in the retentate. Once the retentate becomes highly concentrated and low in volume, further water can be added and repeated (diafiltration). Consequently, we recommend that future studies aim to increase the protein content of the alkaline extraction/ultrafiltration process protein concentrates by more extensive ultrafiltration combined with diafiltration. 

The total mass yield of green coffee protein concentrate prepared by alkaline extraction/ultrafiltration at 19.27 ± 0.30 was approximately double (*p* < 0.05) compared to that prepared by alkaline extraction/acid precipitation. This finding highlights the higher protein losses in the isoelectric precipitation process, where some protein types may not precipitate under acidic conditions (e.g., globulins); however, all proteins should be captured in the retentate of the ultrafiltration process. Similarly, a significantly higher (*p* < 0.05) protein yield was observed for alkaline extracted/ultrafiltered protein concentrate at 64.42 ± 2.48 compared to the alkaline extracted/acid precipitated one (29.85 ± 0.60). 

#### 3.5.2. In Vitro Protein Digestibility (IVPD) of Green Coffee Protein Concentrates Prepared by Alkaline Extraction/Isoelectric Precipitation and Alkaline Extraction/Ultrafiltration 

Table 1 shows the in vitro protein digestibility (IVPD) of the green coffee protein concentrates.

The protein concentrate of green coffee prepared by isoelectric precipitation has significantly (*p* < 0.05) higher digestibility than that prepared by ultrafiltration. A similar result on high protein digestibility of isoelectric precipitated (ca. 90%) lentil proteins compared to ultrafiltration (ca. 78%) was reported by Osemwota et al. [22]. It is possible that the isoelectric precipitation process led to some unfolding of the protein’s secondary and tertiary structure, rendering more hydrolysis sites available for enzyme action [22]. The low digestibility of this protein concentrate and the very low yields indicate that spent coffee may not be a suitable raw material for commercial protein concentrate manufacture.

#### 3.5.3. In Vitro Protein Digestibility (IVPD) of Spent Coffee Protein Concentrate Prepared by Alkaline Extraction/Isoelectric Precipitation

Given the potential of spent coffee to be a low-cost waste utilisation source for protein, we also determined the IVPD of its protein concentrate prepared by alkaline extraction/isoelectric precipitation. Table 1 shows that this protein concentrate had a very low digestibility (*p* < 0.05) compared to the green coffee protein concentrates. This may be due to the denaturation of the proteins and/or the binding of polyphenols to proteins during coffee roasting and brewing.

#### 3.5.4. Amino Acid Profile, Amino Acid Score and In Vitro Protein Digestibility Corrected Amino Acid Score (PDCAAS) and Branched-Chain Amino Acid Data of the Green Coffee Protein Concentrates 

Table 2 gives the profile of essential amino acids, limiting amino acid score, and PDCAAS in green coffee and commercial protein concentrates. The lower the PDCAAS, the more of the protein source would be required to provide enough essential amino acids to support healthy growth or body maintenance. The amino acid scores (based on adult requirements) of the two green coffee protein concentrates show that they are “complete” proteins (score > 1), with slightly lower scores than the commercial soy protein concentrate but similar scores to that of the commercial pea protein concentrate. In the case of the green coffee protein concentrate prepared by isoelectric precipitation, its high digestibility (Table 2) means that its PDCAAS is still greater than 1. However, the lower digestibility of the green coffee protein concentrate prepared by ultrafiltration means that the PDCAAS is far less than 1, indicating it is not “complete”. Neither the coffee protein concentrates nor the commercial soy and pea would be considered “complete” for the needs of a 1-year-old and therefore are not recommended as the main protein source for infant feeding.

In a review of previous literature, Campos-Vega et al. [3] reported that spent coffee grounds had very high levels of essential branched-chain amino acids (BCAA) that may be beneficial to improved exercise performance and more rapid post-exercise recovery. Values of between 21.7–23.0% BCAA of protein for coffee have been reported [23] compared to only 8.9% for soy meal. However, our results (Table 2) do not agree with these previous findings; the coffee protein concentrates had lower levels of BCAA than the commercial soy, and pea protein concentrates.

Protein with a high Fischer ratio (BCAA/AAA) is reported to assist those suffering from malnutrition associated with cancers, burns, trauma, and liver failure, and even may assist in supplementing nutrients to children with chronic or acute diarrhoea or milk protein allergies [3]. Spent coffee proteins have been reported to have a very high Fisher ratio, from 2.6–24.1. Our spent coffee protein concentrate had a Fisher ratio of 1.8 (full data not presented), which was slightly lower than that for green coffee protein concentrates, and the commercial concentrates from soy and pea (Table 2). 

There is very little data in the literature on the amino acid composition of protein from different coffee fractions; therefore, it is still unclear if some coffee protein from some sources, e.g., different varieties or grown in different locations (i.e., genotype x environment), may have high BCAA and Fisher ratio.

#### 3.5.5. Polyphenol Content of the Green Coffee Protein Concentrates and the Spent Coffee Protein Concentrate

Table 1 shows the polyphenol content of the green coffee protein concentrates prepared by alkaline extraction/isoelectric precipitation and alkaline extraction/ultrafiltration and the spent coffee protein concentrate prepared by alkaline extraction/isoelectric precipitation. The two green coffee protein concentrates have similar (*p* > 0.05) polyphenol content. However, the spent coffee protein concentrate has significantly (*p* < 0.05) higher polyphenol content than the two green coffee protein concentrates. The polyphenol content of spent coffee protein concentrate in this study is around two-fold more than that reported by but less than half compared to Samsalee and Sothornvit [24]. A similar finding was found in a study of amaranth flour, where the polyphenol content of protein concentrate was higher than its whole flour; this suggests that the polyphenols may preferentially associate with the protein fraction. Likewise, protein isolates from different plants showed varying levels of polyphenols (in mgGAE/g db) such as whole flour of black bean, 74.99 [25]; sunflower kernel,10.3 [26]; pea 115.5; hemp whole 328.0 and hemp hulled 79.0 [27].

Our findings indicate that these coffee protein concentrates are very high in polyphenols that may provide health benefits above and beyond the protein content due to the oxidative stress protective potential (e.g., antioxidant and anti-inflammatory effects) of these phytochemicals [28]. However, from a negative aspect, the high levels of polyphenols in the spent coffee protein concentrate may have contributed to its low protein digestibility (Table 1). In addition, these polyphenols may give an unacceptable level of bitter or astringent flavours in the protein concentrate. 

### 3.6. Antioxidant Capacity of the Green Coffee Protein Concentrates and the Spent Coffee Protein Concentrate

Table 1 shows the ABTS radical scavenging capacity of the green coffee protein concentrates prepared by alkaline extraction/isoelectric precipitation and alkaline extraction/ultrafiltration and the spent coffee protein concentrate prepared by alkaline extraction/isoelectric precipitation. The protein concentrate of green coffee prepared by alkaline extraction/isoelectric precipitation has a significantly higher (*p* < 0.05) ABTS scavenging capacity than the coffee protein concentrate prepared by alkaline extraction/ultrafiltration. The spent coffee concentrate has an intermediate value (*p* < 0.05). Karaś et al. [29] reported the ABTS scavenging capacity in protein isolates prepared by alkaline extraction/isoelectric precipitation from raw and boiled yellow string beans to be ca. 54% and 51%, respectively. The ABTS scavenging capacity of the spent coffee protein concentrate in this study is around two-fold more than spent coffee grounds [30], but substantially lower than that reported by. The chemical assay we used to assess the antioxidant properties of the samples may not accurately indicate the oxidative stress-protective effects of the extracts in vivo. Therefore, we recommend that cell culture studies be conducted to determine how well the extracts protect against oxidative stress.

## 4. Conclusions

Coffee protein concentrates were successfully manufactured from green coffee beans, roast coffee beans, spent coffee and silver skin by alkaline extraction/isoelectric precipitation. In addition, alkaline extraction/ultrafiltration was also used to prepare a protein concentrate from green coffee beans. The protein content of all concentrates was below the minimum of 60 g/100 g db generally required for a protein concentrate. Therefore, we recommend more extensive concentration by ultrafiltration/diafiltration of green coffee protein extract to increase the protein content. Ultrafiltration/diafiltration of green coffee protein extract gave higher yields than alkaline extraction/isoelectric precipitation indicating the commercial potential of ultrafiltration. High in vitro protein digestibility and a good PDAAS were found for the green coffee alkaline extraction/isoelectric precipitation concentrate, but that prepared by ultrafiltration/diafiltration had lower in vitro protein digestibility and thus lower PDAAS. Increasing the in vitro digestibility of the ultrafiltered green coffee protein concentrate is suggested as a goal for the future. For example, the interesting finding of this research is that the ultrafiltration process gave the highest yield, but lower digestibility. This might be linked to differences in the higher-level protein structures between the ultrafiltration and isoelectric precipitated protein concentrates that need further investigation.

In contrast to some of the literature, we did not observe high levels of BCAA or a high Fischer ratio in the coffee proteins; an anomaly that needs further investigation by analysing the amino acid profiles of coffee beans of different varieties grown in different locations. On the other hand, we observed very high levels of polyphenolics and antioxidant properties in the protein concentrates, an attribute that is highly desirable by consumers. Therefore, with further research and development, we suggest that alkaline extraction with ultrafiltration/diafiltration of green coffee beans could provide a concentrate with good protein content, yields, nutritional quality, and high antioxidant properties. However, its techno-functional properties and sensory acceptability still need investigation.

## Figures and Tables

**Figure 1 foods-11-03244-f001:**
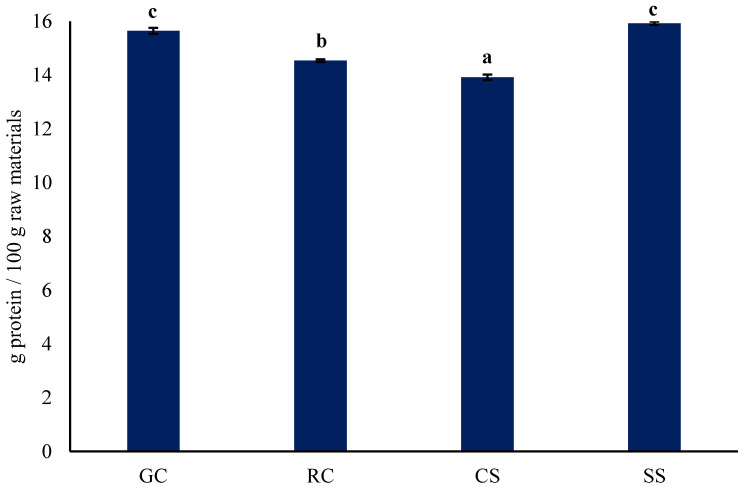
Protein content of coffee fractions before protein extraction (dry basis). GC, green coffee; RC, roasted coffee; CS, spent coffee; SS, coffee silver skin. Mean ± standard deviation (*n* = 2) (dry basis). Means with different letters are significantly different (*p* < 0.05). Protein = *N* × 6.25.

**Figure 2 foods-11-03244-f002:**
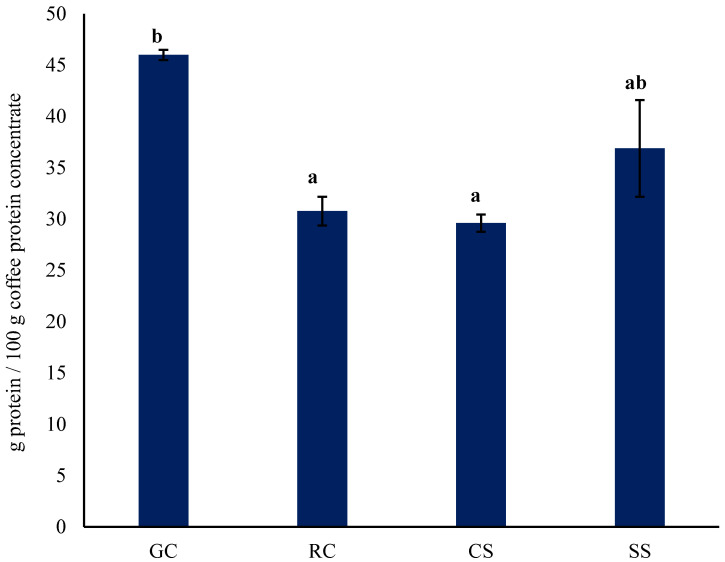
Protein content of protein concentrates prepared by alkaline extraction/isoelectric precipitation (dry basis). GC, green coffee; RC, roasted coffee; CS, spent coffee; SS, coffee silver skin. Mean ± standard deviation (*n* = 2) (dry t basis). Means with different letters are significantly different (*p* < 0.05). Protein = *N* × 6.25.

**Figure 3 foods-11-03244-f003:**
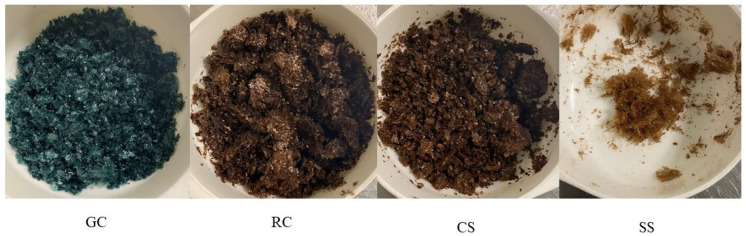
Images of the protein concentrates prepared by alkaline extraction/isoelectric precipitation each from one batch (20 g) of raw material. GC, green coffee; RC, roasted coffee; CS, coffee spent; SS, silver skin.

**Figure 4 foods-11-03244-f004:**
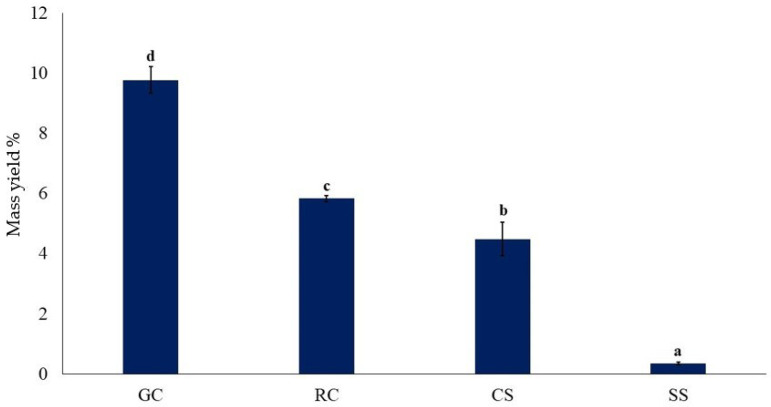
Total mass yields (%) of protein concentrate prepared by alkaline extraction/isoelectric precipitation. GC, green coffee; RC, roasted coffee; CS, spent coffee; SS, coffee silver skin. Mean ± standard deviation (*n* = 2) (dry basis). Means with different letters are significantly different (*p* < 0.05).

**Figure 5 foods-11-03244-f005:**
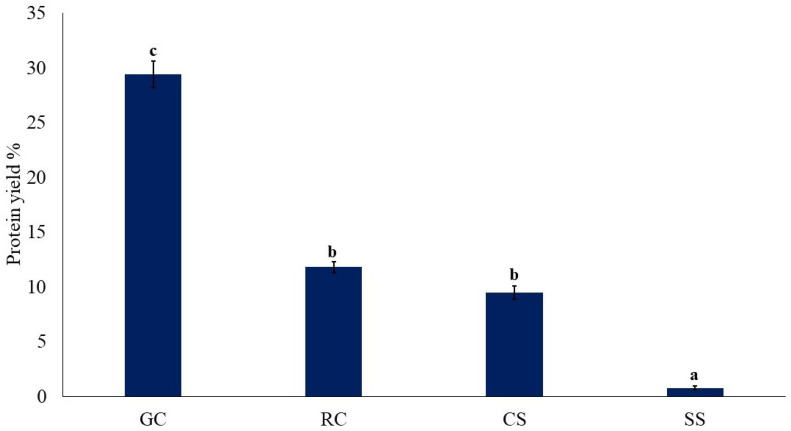
Protein yield of protein concentrates prepared by alkaline extraction/isoelectric precipitation. GC, green coffee; RC, roasted coffee; CS, spent coffee; SS, coffee silver skin. Mean ± standard deviation (*n* = 2) (dry basis). Means with different letters are significantly different (*p* < 0.05).

**Table 1 foods-11-03244-t001:** In vitro protein digestibility (IVPD), phenolic content and antioxidant capacity of selected coffee protein concentrates.

Type of Protein Concentrate	Caffeine Content ^τ^ (g/100 g db)	IVPD (%)	Phenolic Content (mg Gallic Acid Equivalents/g db)	ABTS Scavenging Capacity (mg Trolox Equivalents/g db)
Green coffee protein concentrate prepared by alkaline extraction/isoelectric precipitation	0.16 ± 0.57	96.8 ± 0.86 ^c^	52.6 ± 0.1 ^a^	90.1 ± 2.8 ^c^
Green coffee protein concentrate prepared by alkaline extraction/ultrafiltration	0.10 ± 0.23	50.9 ± 2.4 ^b^	51.0 ± 1.3 ^a^	38.2 ± 2.8 ^a^
Spent coffee protein concentrate prepared by alkaline extraction/isoelectric precipitation	0.16 ± 0.52	33.4 ± 0.3 ^a^	68.6 ± 2.5 ^b^	53.0 ± 0.9 ^b^

Mean ± standard deviation (*n* = 3 **^τ^**, *n* = 2) (dry basis). Different lower case superscript letter within a column indicates a significant difference, *p* < 0.05.

**Table 2 foods-11-03244-t002:** Profile of essential amino acids, limiting amino acid score, PDCAAS and branched-chain amino acid data for the green coffee protein concentrates and two commercial protein concentrates.

Essential Amino Acid	Content mg/g Pure Protein (*N* Conversion Factor for Protein = 6.25)				Human Amino Acid Requirements mg Amino Acids/g Protein ^1^	
	Green Coffee Protein Concentrate Prepared by Isoelectric Precipitation	Green Coffee Protein Concentrate Prepared by Ultrafiltration	Commercial Soy Protein Concentrate ^2^	Typical Pea Protein ^3^	1 Year Old	Adult
Histidine	17 * (1.06) 1.03	24	26	25	26	16
Isoleucine	30	20	48	48	46	13
Leucine	78	49	78	85	93	19
Lysine	41	20 * (1.25) 0.64	63	75	66	16
Threonine	30	20	39	37	43	9
Tryptophan	ND	ND	14	10	17	5
Valine	43	26	48	50	55	13
Methionine + cysteine	34	26	27 * (1.59)	19 * (1.10)	42	17
Tyrosine + phenylalanine	78	46	88	55	72	19
BCAA	15.1	9.5	17.4	18.3		
AAA	7.8	4.6	8.8	5.5		
Fischer ratio	2.4	2.1	2.0	3.3		

PDCAAS, protein digestibility corrected amino acid score. ^1^ FAO (1991). ^2^ Arcon, SMB soy protein concentrate (ADM). ^3^ https://vegfaqs.com/pea-protein-amino-acid-profile/. * Indicates the limiting amino acid, followed by the amino acid score for adults in brackets and the PDCAAS for adults in bold text. ND, not determined. BCAA (Val + Leu + Ile as % of pure protein), AAA (Phe + Tyr as % of pure protein), Fischer ratio (BCAA/AAA) [3].

## Data Availability

The datasets generated for this study are available on request to the corresponding author.

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
