# Peer review of "Extraction, Isolation and Nutritional Quality of Coffee Protein"

_foods, 2022, doi:10.3390/foods11203244_

Round 1

Reviewer 1 Report

The work presented by Bhattarai et al. describes the application of isoelectric precipitation and ultrafiltration to the alkaline protein extracts of coffee, with the aim of obtaining protein concentrates. The work is well described, and the methodologies are adequate.

Lines 65-66: there is a recent study in which enzyme-assisted extraction was applied to recover a fraction rich in proteins and polyphenols from by-products of the coffee industry, tested for antimicrobial, antithyrosinase and antioxidant activity (Prandi B, Ferri M, Monari S, Zurlini C, Cigognini I, Verstringe S, Schaller D, Walter M, Navarini L, Tassoni A, Sforza S, Tedeschi T. Extraction and Chemical Characterization of Functional Phenols and Proteins from Coffee (Coffea arabica) By-Products. Biomolecules. 2021 Oct 22;11(11):1571. doi: 10.3390/biom11111571. PMID: 34827569; PMCID: PMC8615506.)

Lines 80-82: why were soy proteins chosen as a reference and not others?

Lines 100-103: what was the amount of non-protein nitrogen (e.g., caffeine residues) in the protein concentrate? Or, since you have the amino acid composition, you can calculate the exact conversion factor instead of using 6.25.

Lines 153-156: table 2 is named before table 1.

Table 2: was the tryptophan not determined or was it not present?

Lines 180-181: why was the ABTS method used and not for example the DPPH method?

Figure 1: in my opinion these values could be overestimated due to the presence of caffeine.

Lines 195-196: caffeine could also affect these results.

Figure 1 and 2: to determine only protein nitrogen, it would be better to graph these with the total amount of amino acids, to exclude all non-protein nitrogen compounds.

Figure 2: proteins are between 30 and 45% of the dry extract, what is the remaining part?

Figure 4: both the protein purity obtained with alkaline extraction/acid precipitation and the yield are low, is the process sustainable from an economic point of view?

Lines 279-281: what is the remaining 50%? At least a centesimal analysis should be done. Since it hasn't been eliminated by ultrafiltration, it looks like something that weighs more than 5 kDa, but it should be identified (at least the category).

Lines 355-357: loss of formatting. From here, no more line numbering.

Paragraph 3.5: you are comparing the results obtained on the ABTS test with the literature on DPPH tests. These may not be directly comparable. Please refer to the same essay.

Conclusions: the claims should be less enthusiastic, the protein yield is not high, as is the protein purity, and the technique that gives the highest yield is unfortunately the one that gave extracts with the lowest digestibility. Thus, the work is a good starting point for studying the production of coffee protein concentrates, but there is still a lot of work to be done.

Author Response

Thank you very much for your constructive feedback. We have now incorporated your suggestions into the manuscript.

Reviewer 2 Report

The purpose of this article is to perform the extraction and analysis of coffee proteins. One major concern is the authors did not perform any experiments to study the purity of coffee proteins. The authors mentioned several time that one extraction method provides higher yield and purity. How to determine the purity? Why not perform gel electrophoresis to study the protein profile? Please find my other comments below.

L17-18: What are other fractions?

L137: What is the enzyme activity for pancreatin?

L193: This statement is very confusing. How to determine protein content without protein extraction?

Suggest the authors to change to y-axis in Figure 1 and 2 to g protein/100 g raw materials and g protein/100 g coffee protein concentrate.

L199-200: I did not see any relevant information about protein content from Table 1.

L220-221: How did the authors know about the purity? Based on the comparison among different plant proteins, the lower protein concentration does not equal to the lower purity.

L235-236: Why not perform color measurement (L, a, b) to compare the color difference?

L241-243: What are the compounds responsible for the color?

What is the difference between Figure 2 and Figure 5? Why silver skin coffee has such a smaller protein yield but leads to a high protein content? It does not make sense.

L279  and L291: Why the total mass yield doubled but protein content was not doubled?

Author Response

(The authors gave the same response as above.)

Round 2

Reviewer 2 Report

Authors have addressed the previous comments.